# Optimization of mine ventilation network feature graph

**Jinzhang Jia**[1,2], **Bin Li**[1,2]*, **Dinglin Ke**[1,2], **Yumo Wu**[3], **Dan Zhao**[3], **Mingyu Wang**[4]

**1** College of Safety Science and Engineering, Liaoning Technical University, Fuxin, Liaoning, China, **2** Key Laboratory of Mine Power Disaster and Prevention of Ministry of Education, Huludao, Liaoning, China, **3** College of Civil Engineering, Guangdong Ocean University Cunjin College, Zhanjiang, Guangdong, China, **4** Power China Hua Dong Engineering Corporation Limited, Zhejiang, Hangzhou, China

* lb471910086@163.com

**Data Availability Statement:** All relevant data are within the paper and its Supporting Information files.

**Funding:** This research was supported by the National Natural Science Foundation of China (No. 51374121) and funded by Liaoning Distinguished

## Abstract

A ventilation network feature graph can directly and quantitatively represent the features of a ventilation network. To ensure the stability of airflow in a mine and improve ventilation system analysis, we propose a new algorithm to draw ventilation network feature graphs. The independent path method serves as the algorithm's main frame, and an improved adaptive genetic algorithm is embedded so that the graph may be drawn better. A mathematical model based on the node adjacency matrix method for unidirectional circuit discrimination is constructed as the drawing algorithm may not be valid in such cases. By modifying the edge-seeking strategy, the improved depth-first search algorithm can be used to determine all of the paths in the ventilation network with unidirectional circuits, and the equivalent transformation method of network topology relations is used to draw the ventilation network feature graph. Through the analysis of the topological relation of a ventilation network, a simplified mathematical model is constructed, and network simplification technology makes the drawing concise and hierarchical. The rapid and intuitive drawing of the ventilation network feature graphs is significant for optimization of the ventilation system and day-to-day management.

## 1. Introduction

With the increase of mining depths and the expansion of mining's scale [1], mine ventilation systems are becoming larger and their topology more complex, which makes their simulation, optimization, and prediction more difficult [2, 3]. To meet the needs of modern, scientific management of ventilation systems, optimization of the drawing methods of complex ventilation networks is necessary. The ventilation network feature graph (also known as a Q-H graph) is a new way to directly and quantitatively represent the state of a ventilation system, and is an effective tool to research complex ventilation networks [4].

Euler established graph theory in the 18th century with the study of the Königsberg seven-bridge problem. In the 19th century, Guthrie made the four-color conjecture, which became the driving force in the development of graph theory [5, 6]. In the 20th century, the random graph theory of Erdos and Renyi was a great theoretical leap [7]. In recent years, the research results of two-dimensional graphs have been widely applied to draw mine ventilation network

Professor (551710007007), funded project of the Liaoning Million Talents project (2019-45-15), and the Natural Science Foundation of Liaoning Province (2019-MS-162). The funders had no role in study design, data collection and analysis, decision to publish, or preparation of the manuscript. The commercial company ' Power China Hua Dong Engineering Corporation Limited' provided support in the form of salaries for the author Mingyu. Wang, but did not have any additional role in the study design, data collection and analysis, decision to publish, or preparation of the manuscript.

**Competing interests:** The authors have declared that no competing interests exist. The commercial company ' Power China Hua Dong Engineering Corporation Limited' provided support in the form of salaries for the author Mingyu. Wang, but did not have any additional role in the study design, data collection and analysis, decision to publish, or preparation of the manuscript. The commercial company ' Power China Hua Dong Engineering Corporation Limited' does not provide experimental funds and will not use the relevant information of the paper to develop products and apply for patents, and will not mind appearing as an author unit. This does not alter our adherence to PLOS ONE policies on sharing data and materials.

graphs [8]. Mine ventilation system graphs and network graphs currently see much study, resulting in remarkable progress in algorithm optimization, topology transformation, and effect visualization. Deng et al. [9] introduced the stratification method into the ventilation network drawing, and to divide the ventilation network into different levels by using "node stratification", which reduced the difficulty of solving the minimum problem of branch intersection and improved the drawing effect; Wei et al. [10] studied the topology structure characteristics of the complex ventilation system network and the process and principle of mutual transformation, and applied the research results to the drawing of the ventilation network in Sanhejian Coal mine, which proved that the results effectively improved the drawing speed; based on the idea of graph theory, Wang et al. [11] proposed the algorithm of numbering and weight value of branch nodes, and realized the stereo visualization effect of ventilation system graph. However, these graphs can only reflect topological relationships of a ventilation system, and not their quantitative relationships. The Q-H graph can effectively solve this problem. In a Q-H graph, each branch in a ventilation network is represented by a rectangular block whose width, height, and area respectively equal the wind volume, resistance, and power consumption of the branch. The rectangular blocks are arranged in accordance with the topological relation of the network to form a wind resistance equilibrium graph [4]. The Q-H graph is equivalent to the network graph, which reflects not only the relation between nodes, circuits, and branches but the air quantity, resistance, and power consumption of each branch. It is more intuitive and quantitative than the network graph [12]. Huang. [13] preliminarily studied the essence and application of the Q-H graph, but did not mention its drawing algorithm. Xu et al. [12, 14] proposed the "four-line" feature of the Q-H graph, that is, the function of the node, circuit, cut set, and path lines. A study was conducted on the Q-H graph drawing algorithm. Exhaustive search was used for networks, and all rectangular blocks were put together according to the search process to form the Q-H graph of a ventilation network. Based on the "four-line" feature analysis of the Q-H graph. Zhou et al. [15] put forward the resistance reduction measures of the gas extraction system, which significantly improved the gas extraction effect of the drilling field. Sun [16] analyzed the relationship between underground air leakage and spontaneous combustion of coal by the Q-H graph, determined the control range of air leakage, and worked out an optimization scheme. Through the verification of practical engineering, the air leakage in goaf decreased from $750m^3$/min to $150m^3$/min. This shows that it is feasible to control air leakage by using the Q-H graph method. Although the above researches promote the application and development of the Q-H graph, there is a lack of in-depth research on the algorithm of the Q-H graph drawing.

The key of the existing drawing method in [14] is to turn the depth-first search method of the artificial intelligence into the exhaustive method, and Q-H graph can be formed by putting all the rectangular blocks together according to the search process. As shown in Fig 1, the search process [14] is:$e_1 \rightarrow e_3 \rightarrow e_7 \rightarrow (e_3) \rightarrow e_8 \rightarrow e_9 \rightarrow (e_8 \rightarrow e_3 \rightarrow e_1) \rightarrow e_4 \rightarrow e_5 \rightarrow (e_4)$ $\rightarrow e_6 \rightarrow (e_4 \rightarrow e_1) \rightarrow e_2$ (The procedure in parentheses is backtracking). This paper based on the idea of independent path method (IPM), the depth-first search method is used to determine all of the independent paths. Then a Q-H graph can be drawn by establishing the coordinates of the lower-left and upper-right corners of the rectangular blocks corresponding to the branches of the independent paths. As shown in Fig 1, a total of 3 independent paths were found, namely $P_1=\{e_1,e_3,e_7\}$, $P_2=\{e_1,e_4,e_5,e_8,e_9\}$, $P_3=\{e_2,e_6,e_9\}$.For large-scale ventilation networks, the drawing method in [14] will waste a lot of time.

Based on the drawing model of the independent path method, this paper uses an adaptive and improved genetic algorithm to optimize the sorting problem of independent paths in order to solve the problem of the large number of divided rectangular blocks in the drawing process, so that the number of rectangular blocks divided in the QH graph is reduced. and the

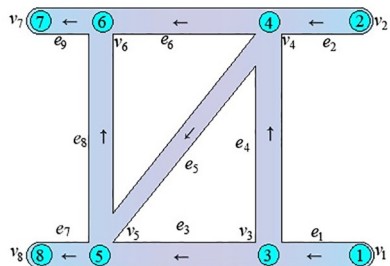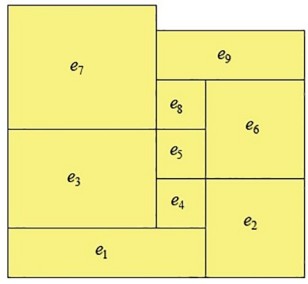

**Fig 1. Transformation of network graph and Q-H graph in [14].** (a) Ventilation network graph; (b) Q-H graph.

drawing effect is more intuitive. A mathematical model based on the node adjacency matrix method for unidirectional circuit discrimination is constructed so that the unidirectional circuit in ventilation network can be accurately identified. By modifying the edge-seeking strategy, the improved depth-first search algorithm can be used to determine all of the paths in the ventilation network with unidirectional circuits, and the equivalent transformation method of network topology relations is used to draw the ventilation network with unidirectional circuits feature graph. In addition, the application of network equivalent automatic simplification technology in Q-H graph drawing can shorten the drawing time and make the Q-H graph hierarchical. The powerful function of Q-H graph can provide a theoretical basis and technical support to intelligent diagnosis, reliability analysis, and optimization of complex ventilation network systems. It can be applied to ventilation systems such as of subways, tunnels, and large shopping malls.

## 2. Drawing model and optimization research of Q-H graph

When building a mathematical model of a Q-H graph drawing based on the independent path method, the depth-first search method is used to determine all of the independent paths. Then the coordinates of the lower-left and upper-right corners of the rectangular blocks corresponding to the branches of the independent paths are established. This is suitable for various network models.

### 2.1 Mathematical model of Q-H graph drawing based on independent path method

A ventilation network graph is denoted as $G = (V,E)$, where $V = \{v_1,v_2,...,v_m\}$ is the set of m nodes, and m is usually written as $m = |V|$. E is the set of $n$ branches.

To draw the rectangular block corresponding to the ventilation network branch $e_k = (v_i,v_j)$ on the graph, the coordinates of the points at the lower-left corner $(x_k,y_k)$ and upper-right corner $(x'_k, y'_k)$ must be determined. The Q-H graph drawing process based on the independent path method is as follows:

(1) Determine the ordinate of the network nodes

If the pressure energy $H_i$ of node $v_i$ is known, which is regarded as the corresponding ordinate of Q-H graph, and the ordinate of its adjacent node is:

$$H_j = \begin{cases} H_i + h_k & (e_k = (v_i, v_j) \in E) \\ H_i - h_k & (e_k = (v_j, v_i) \in E) \end{cases} \tag{1}$$

where $h_k$ is the resistance of branch $e_k$, Ns$^2$/m$^8$. Take any node of the network (generally the

node with the highest pressure energy) as the base point, and define its ordinate as 0. According to the above rules, the ordinates of all of the nodes can be determined.

(2) Use the depth-first algorithm [17] to search the path from source to intersection.

The edge-seeking strategy is weighted by the value of the minimum air quantity of the branch. If the starting node of the edge seeking is $v_a$, then the edge-seeking strategy is:

$$e_k = \{(v_a, v_b)|(v_a, v_b) \in E, q_{ab} > \varepsilon, q_k = \min(q_{ab})\} \tag{2}$$

where $q_{ab}$ is the air quantity corresponding to branch $(v_a, v_b)$, m³/s; $q_k$ is the air quantity corresponding to branch $e_k$, m³/s; and $\varepsilon$ is the bound on the error of the air quantity.

For planar networks, the left and right order of network branches can be used as the edge-seeking strategy. A Q-H graph drawn in this way is consistent in position with the network graph, but the stereo network must take formula (2) as the edge-seeking strategy.

The first path determined by the search is:

$$\begin{aligned}
P_I &= \{P_I[1], P_I[2], \ldots, P_I[j], \ldots, P_I[|P_I|]\} \\
&= \{e'_1, e'_2, \ldots, e'_j, \ldots, e'_{|P_I|}\}
\end{aligned} \tag{3}$$

where $P_I[j]$ and $e'_j$ are the two ways of writing the j th branch of path $P_I$, j=1,2,. . .,$|P_I|$.

(3) Define the air quantity of the branch with the minimum air quantity in the path as the width of the path.

The width of path $P_I$ is:

$$w_I = \min\{q'_1, q'_2, q'_3, \ldots, q'_j, \ldots, q'_{|P_I|}\} \tag{4}$$

where $q'_j$ is the air quantity corresponding to the j th branch $e'_j$ of path $P_I$, m³/s.

(4) "Color" the path branches [17].

The air quantity of each branch in the path subtracts the width of the path. This is regarded as the branch "coloring" of the path, that is:

$$q'_j \leftarrow q'_j - w_I, (e'_j \in P_I) \tag{5}$$

At this point in the first iteration, one path has been identified. We return to the search source and identify the next path, and continue for a total of (n-m+2) paths.

(5) Perform intersection operation on all of the paths.

Intersection is performed between the current path $P_I$ and previous path $P_{I-1}$. Let:

$$\begin{cases}
E'' = P_{I-1} - P_{I-1} * P_I = (e''_1, e''_2, \ldots, e''_{|E''|}) \\
E''' = P_{I-1} - P_{I-1} * P_I = (e'''_1, e'''_2, \ldots, e'''_{|E'''|})
\end{cases} \tag{6}$$

(6) Determine the coordinates of the point at the lower-left corner of the rectangular block.

(i) The branch belonging to the first path, $P_1$, has coordinates

$$\begin{cases}
x_k = 0 \\
y_k = H_i
\end{cases} \{e_k = (v_i, v_j) \in P_1\} \tag{7}$$

(ii) The branch belonging to $E'''$ has coordinates

$$\begin{cases} x_k = \sum_{s=1}^{I-1} w_s & \{e_k = (v_i, v_j) \in E'''\} \\ y_k = H_i \end{cases} \tag{8}$$

(7) Determine the coordinates of the upper-right corner of the rectangular block.
(i) The branch belonging to $E''$ has coordinates

$$\begin{cases} x'_k = \sum_{s=1}^{I-1} w_s & \{e_k = (v_i, v_j) \in E''\} \\ y'_k = H_j \end{cases} \tag{9}$$

(ii) The branch belonging to the current path $P_I$, whose air quantity is less than $\varepsilon$, has coordinates

$$\begin{cases} x'_k = \sum_{s=1}^{I} w_s & \{e_k = (v_i, v_j) \in E'', q_k \leq \varepsilon\} \\ y'_k = H_j \end{cases} \tag{10}$$

(8) Repeat the above steps until all branch coordinates are determined. The above process is a form of determining independent paths, and therefore we call this the independent path method.

This method is suitable for drawing Q-H graphs of both planar and stereo networks. A planar network is shown in Fig 2(A), with network parameters as shown in Table 1. There is no crossover between branches except nodes in the network graph, and the left and right sequences of network branches are arranged as branch weights in the edge-seeking strategy. In this way, the Q-H graph is consistent in position with the network graph, and the rectangular blocks corresponding to branches are not divided, as shown in Fig 2(B). The Q-H graph of a stereo network will have rectangular blocks corresponding to some branches divided into two or more blocks, regardless of the edge-seeking strategy. Research shows that the number of rectangular blocks after division is closely related to the sequence of independent paths, as

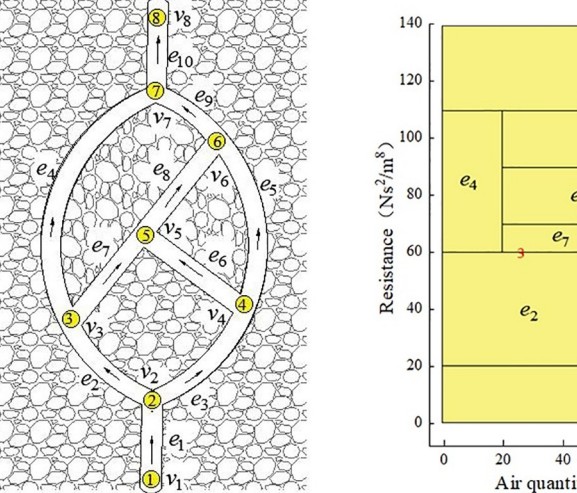

**Fig 2. Transformation of network graph and Q-H graph.** (a)Ventilation network graph;(b)Q-H graph.

**Table 1. Network parameters.**

| Branch | $e_1$ | $e_2$ | $e_3$ | $e_4$ | $e_5$ | $e_6$ | $e_7$ | $e_8$ | $e_9$ | $e_{10}$ |
|---|---|---|---|---|---|---|---|---|---|---|
| Air quantity($m^3$/s) | 100 | 60 | 40 | 20 | 30 | 10 | 40 | 50 | 80 | 100 |
| Resistance($Ns^2/m^8$) | 20 | 40 | 20 | 50 | 50 | 30 | 10 | 20 | 20 | 30 |

determined by the search strategy in formula (2). Obtaining the optimal sequence of independent paths is a focus of this paper.

## 2.2 Optimization of Q-H graph segmentation based on improved adaptive genetic algorithm

To obtain an optimal independent path sequence to reduce the rectangular block cutting in a Q-H graph, we apply the improved adaptive genetic algorithm (IAGA) to optimize the independent path sequencing problem. Each topological relation change that determines the sequence of the independent paths is taken as a solution in the solution space, and IAGA is used to find the optimal or approximate optimal solution by a random search in the solution space, so as to reduce the workload of the path comparison sequence. The process of Q-H graph optimization based on IAGA is described next.

**2.2.1 Ventilation network topology transformation coding.** Hybrid binary and integer encoding based on node topology transformation is adopted. When the nodes of a mine ventilation network are coded, we must consider node output degrees $|E^+(V_i)|$ of 1, 2, and 3, as shown in Fig 3.

With output degree 1, only one node is connected, with no change of topological relationship, so there is no coding.

An output degree of 2 can be represented by bit binary encoding, where 0 and 1 represent different topological sequences of the two branches connected to the node.

For output degree 3, branch topologies are represented by a series of integers. The number of topological sorting variations of the three branches is $|E^+(V_i)|! = 3 \times 2 \times 1 = 6$, corresponding to (1,2,3,4,5,6), respectively, so as to realize a one-to-one correspondence between the coding and solution spaces.

The above mixed encoding method can represent the different independent path sequences found in a Q-H graph due to the change of topological relations.

**2.2.2 Design of fitness function in Q-H graph drawing.** The number of rectangular blocks in a Q-H graph can be expressed as:

$$
\begin{aligned}
K &= |P_1| + (|P_2| - |P_1 \cap P_2|) + (|P_3| - |P_2 \cap P_3|) \\
&\quad + \cdots + (|P_{n-m+2}| - |P_{n-m+1} \cap P_{n-m+2}|) \\
&= |P_1| + \sum_{i=2}^{n-m+2} (|P_i| - |P_i \cap P_{i-1}|)
\end{aligned}
\tag{11}
$$

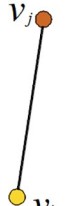 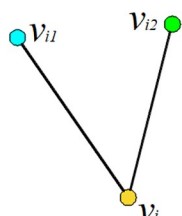 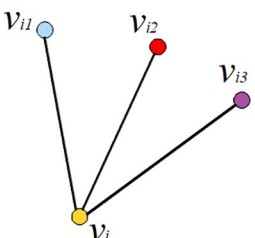

**Fig 3. Different node output degrees.**

The problem of reducing the number of cut rectangles in the Q-H graph drawing is to minimize the objective function of the number of cut rectangles representing the branches. Thus:

$$g(x) = \min(x) = \min(K)$$
$$= \min\left[|P_1| + \sum_{i=2}^{n-m+2}(|P_i| - |P_i \cap P_{i-1}|)\right] \tag{12}$$

where x is the number of rectangular blocks in the Q-H graph.

For the minimization problem, the objective function should be converted to a fitness function when applying a genetic algorithm to calculate fitness, i.e. [18],

$$F(x) = C_{\max} - g(x) \quad if \quad g(x) < C_{\max} \tag{13}$$

where F(x) is the fitness after conversion; g(x) is the fitness under the minimum value problem; that is, the number of rectangular blocks in the Q-H graph; and $C_{\max}$ is a sufficiently large number whose value is determined according to the parameters of the mine ventilation network.

A more accurate expression of the fitness function would be $F(A_i^t)$, which is the fitness value of each chromosome at generation t. The fitness value of each string in the population is calculated, and this can evaluate the optimization degree of any q-dimensional vector in the chromosome string space. In each generation of chromosomes, the number of rectangular blocks in the Q-H graph can be obtained through the Q-H graph, reflecting the performance of each generation of chromosomes and the Q-H graph. Through genetic operations, such as selection, crossover and mutation, the chromosomes are developed towards the regions with better performance, and the chromosomes that reduce the cutting of rectangular blocks in the Q-H graph are finally obtained, which is the optimal solution required to solve the problem.

**2.2.3 Improved adaptive crossover probability $P_c$ and mutation probability $P_m$.** Based on research of the improved genetic algorithm [19–22], we propose an improved adaptive genetic algorithm (IAGA). The larger fitness value of the two intersecting individuals is adaptively adjusted between the minimum $f_{\min}$, average $f_{avg}$, and maximum $f_{\max}$, and the crossover probability $P_c$ and mutation probability $P_m$ are adjusted between the minimum, average, and maximum fitness values of the population with the change of individual fitness value. Thus, according to Lagrange interpolation, we obtain the improved crossover probability,

$$P_c = \frac{(f' - f_{avg})(f' - f_{\max})}{(f_{\min} - f_{avg})(f_{\min} - f_{\max})}P_{c1}$$
$$+ \frac{(f' - f_{\min})(f' - f_{\max})}{(f_{avg} - f_{\min})(f_{avg} - f_{\max})}P_{c2} \tag{14}$$
$$+ \frac{(f' - f_{\min})(f' - f_{avg})}{(f_{\max} - f_{\min})(f_{\max} - f_{avg})}P_{c3}$$

and improved mutation probability,

$$P_m = \frac{(f - f_{avg})(f - f_{\max})}{(f_{\min} - f_{avg})(f_{\min} - f_{\max})}P_{m1}$$
$$+ \frac{(f - f_{\min})(f - f_{\max})}{(f_{avg} - f_{\min})(f_{avg} - f_{\max})}P_{m2} \tag{15}$$
$$+ \frac{(f - f_{\min})(f - f_{avg})}{(f_{\max} - f_{\min})(f_{\max} - f_{avg})}P_{m3}$$

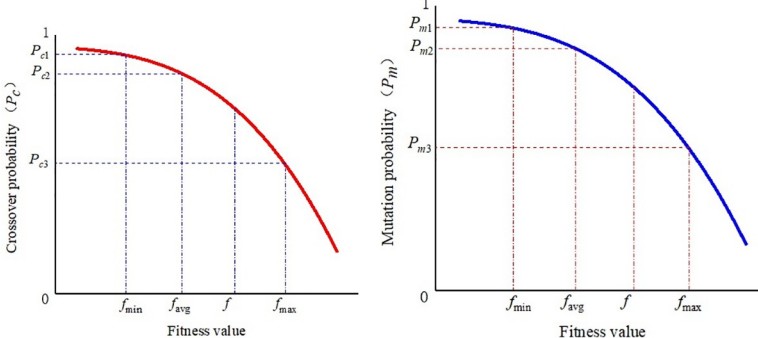

**Fig 4. The change of crossover probability and mutation probability of adaptive genetic algorithm.** (a)Variation of crossover probability $P_c$; (b)Variation of mutation probability $P_m$.

where $f_{max}$ is the maximum fitness value in the population; $f_{avg}$ is the average fitness value of each generation of the population; $f_{min}$ is the minimum fitness value in the population; $f'$ is the larger fitness value of the two intersecting individuals; and f is the fitness value of the individual to be mutated. Moreover, the crossover probability $P_c$ and mutation probability $P_m$ take the value of the interval (0,1), which can be adjusted during the optimization process and $P_{c1} > P_{c2} > P_{c3}$, $P_{m1} > P_{m2} > P_{m3}$.

According to formulas (14) and (15), the variation of crossover probability $P_c$ and mutation probability $P_m$ is obtained as shown in Fig 4.

The improved crossover probability and mutation probability not only can automatically change with the fitness value but ensure that the crossover probability of maximum fitness of individuals in the population is nonzero. The high precision of the adjustment improves the crossover and mutation probabilities of individuals so that they will not be in a stagnant state; hence, the algorithm will jump out of local optimal solutions.

Taking a ventilation network with m = 82 and n = 123 as an example, with other parameters remaining the same, a simple genetic algorithm (GA) and IAGA are used to optimize the Q-H graph drawing. The trends of the best fitness value and average fitness value in each generation are shown, respectively, in Fig 5.

As shown in Fig 5(A), when the Q-H graph is drawn by the GA, the trends of the best and average fitness values in each generation show jumping states, with fewer evolutionary generations and faster convergence. However, in Fig 5(B) when IAGA is used, the trends of the best and average fitness values in all of the generations are stable, and their evolution is obviously better. This indicates that when IAGA is used to draw the Q-H graph, the chromosome

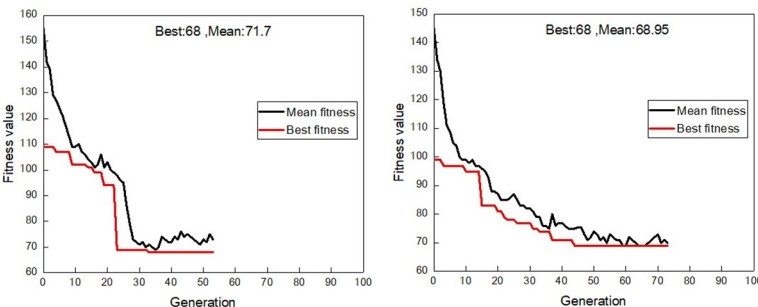

**Fig 5. General genetic algorithm and improved genetic algorithm fitness value changes.** (a)Fitness value variation for each generation by GA;(b) Fitness value variation for each generation by IAGA.

gradually shows a better state with the change of crossover and mutation probabilities in the evolution process of each generation, and a better independent path sequence for Q-H graph drawing can be found more quickly.

**2.2.4 Determination of genetic operators.**   (1) Design of operator selection

In this study, the selection adopts the random traversal sampling method based on the sorting fitness allocation [18] to allocate the selection probability. The method uses the fitness value of the population and sorts by the target value.

Fitness calculations are based only on the order Pos of the individuals in the population, and not the actual target value. Suppose L is the number of individuals in the population, SP is the selection pressure, and the fitness value of each individual is calculated according to its order Pos in the sorted population. The fitness value is calculated as [18]:

$$Fit(Pos) = 2 - SP + \frac{2(SP - 1)(Pos - 1)}{L - 1} \tag{16}$$

where Fit is the fitness value. We sort by the value of Fit, and the minimum fitness individual is placed at the first position in the sorted list of values of the objective function.

Individual selection probability is calculated according to the linear sorting formula proposed by Baker [18]:

$$P_i = \frac{1}{L}\left[\eta^+ - (\eta^+ - \eta^-)\frac{i - 1}{L - 1}\right] \tag{17}$$

where i is the individual serial number, $1 \leq \eta^+ \leq 2$, $\eta^- = 2 - \eta^+$.

According to the calculated selection probability, random traversal sampling is conducted to select individuals, where Fit is a column vector whose value is calculated by formula (16). We set npointer as the number to be selected, and select individuals at equal distances 1/npointer. The position of the first pointer is determined by a uniform random number in the interval [0,1/npointer].

The selection process is as follows: first, the individuals are ranked based on fitness, then the selection probability is calculated, and finally the selection operation of random traversal is completed.

(2) Design of crossover operator

In this paper, the crossover operator adopts the uniform crossover mode and takes every point as a potential rendezvous point. The crossover probability is adjusted adaptively according to IAGA, and the crossover operation is carried out on individuals in the population. Crossover is realized by masking words, randomly generating a (0-1) masking word $W = w_1 w_2 \ldots w_q$ with the same length as the individual encoding length, and producing new individuals A' and B' from parent generation A and B by the following rules: If $W_i = 0$, then the gene values at the ith locus of A' and B' inherit the corresponding gene values of A and B, respectively. If $W_i = 1$, then the gene values at the ith locus of A' and B' inherit the corresponding gene values of B and A, respectively. In other words, the same genes from the father generation are preserved, whereas the offspring inherit some different gene values from the two fathers, which is conducive to improving the quality of the offspring.

For example, if

| Father *A* | 0 | 1 | 0 | 2 | 5 | 4 | 1 |
|---|---|---|---|---|---|---|---|
| Father *B* | 1 | 0 | 1 | 6 | 3 | 2 | 0 |

Sample of masking word:

| Sample 1 | 0 | 1 | 0 | 1 | 0 | 0 | 1 |
|----------|---|---|---|---|---|---|---|
| Sample 2 | 1 | 0 | 1 | 0 | 1 | 1 | 0 |

Two offspring generated after crossover:

| Offspring $A$ | 1 | 1 | 1 | 2 | 3 | 2 | 1 |
|---------------|---|---|---|---|---|---|---|
| Offspring $B$ | 0 | 0 | 0 | 6 | 5 | 4 | 0 |

It can be seen from the uniform crossover process that the new generation's chromosomes after crossing are still a series of integers that can represent the sequence of independent paths of a ventilation network based on node order.

(3) Design of mutation operator

For the binary coding of the Q-H graph drawing algorithm, the mutation operation is to reverse the gene value on the chromosome according to the adaptive mutation probability, that is, 0 to 1 and 1 to 0. For integer coding, the gene value on the chromosome is determined as other value which is not itself according to the adaptive mutation probability.

**2.2.5 Selection of relevant parameters.**   The parameter selection of GA affects the result of Q-H graph drawing. As the mutation and crossover probabilities change adaptively and the selection probability is obtained by linear sorting, the main parameters affecting the algorithm's result are the initial population size and preset maximum of generations [23].

(1) Selection of maximum of generations

Loop termination conditions are provided by presetting the maximum of generations, which depends on the size of the ventilation network. For small ventilation networks, there is no need to set too large a number of generations because there are few changes of chromosome strings. For large ventilation networks, there are many changes in the chromosome string, so it is necessary to set a larger number of generations to find a better solution as soon as possible.

(2) Determination of initial population size

The initial population size generally affects the convergence and computational efficiency of a genetic algorithm. If the scale is too small, then it tends to converge to a local optimal solution. If the scale is too large, then the calculation speed will be reduced [24]. In the Q-H graph drawing optimization algorithm, the initial population depends not only on the number of nodes in the ventilation network, but also on the topological relation changes of nodes in the ventilation network, because the hybrid binary and integer encoding based on node topology transformation is adopted. There are n-m+2 independent paths in the ventilation network, and the number of topological relation changes of the n-m+2 independent paths is (n-m+2)!, the initial population size is less than (n-m+2)!For a specific mine ventilation network, the setting of initial population depends on the specific parameters of it.

## 2.3 Program design of Q-H graph drawing

As shown in Fig 6, steps of Q-H graph optimization drawing based on the hybrid algorithm (IAGA-IPM) combing the improved adaptive genetic algorithm and the independent path method are as follows:

Step1: Determine the basic parameters of the IAGA (population size, maximum of evolutionary generation).

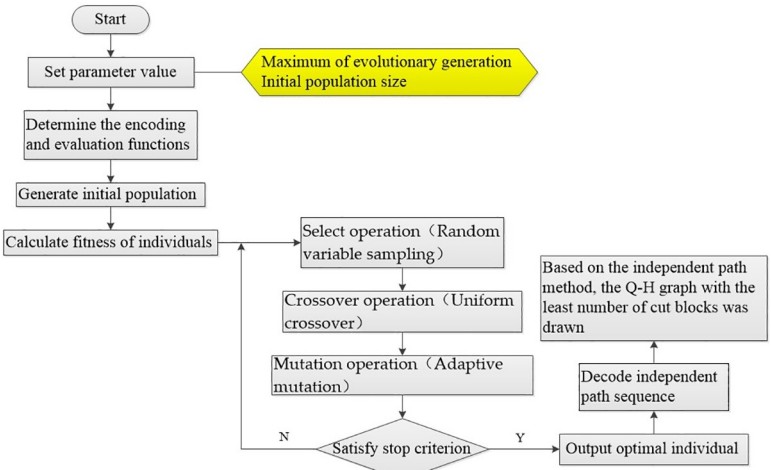

**Fig 6. Q-H graph optimization flowchart based on IAGA-IPM.**

Step2: Select an appropriate encoding method. We adopt a hybrid of binary and integer encoding based on the node to map solution variables (topological relation variations) in the solution space to chromosome strings of IAGA. The objective function of the Q-H graph drawing algorithm optimization problem (reduction of Q-H graph rectangular block cutting) is defined, and the solution variable of the problem to be solved is determined, which represents the change of the topology relation of nodes in the independent path sequence.

Step3: Produce the initial population. Set tt=0 (tt represents the maximum of evolutionary generation), and select the initial population, and calculate the fitness based on sorting.

Step4: Individuals are selected by random sampling, and the crossover parent generations are selected based on selection operator.

Step5: The selected parent generation is performed crossover operation according to the improved adaptive crossover probability $P_c$ using a uniform crossover operator to obtain new individual.

Step6: The new individual generated by crossover is preformed mutation operation based on the improved adaptive mutation probability $P_m$ using the single point mutation operator to produce the new generation of individuals.

Step7: If the maximum of evolutionary generation meets the set requirement, then output the optimal solution, and decode the chromosome string with maximum fitness into the order of (n−m+2) independent path; otherwise, turn to Step4, keep looping.

Step8: Draw the Q-H graph of the mine according to the order of the independent path from Step7 based on the way of independent path.

At this point, the optimal mine Q-H graph can be drawn according to the above steps.

According to IAGA-IPM, the main program and related subroutines of the Q-H graph drawing optimization algorithm are coded, and the visualization effect of Q-H graph drawing is realized. We will test several ventilation networks.

Take the stereo ventilation network with parameters m = 8 and n = 11 as an example, as shown in Fig 7. Based on the convergence situation of multiple runs and the quality of

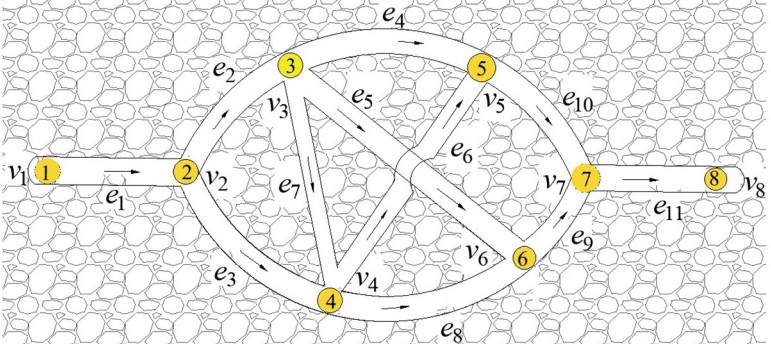

**Fig 7. A stereo ventilation network graph.**

solution, the initial population can be set as 10, and the maximum of evolutionary generations as 50.

The Q-H graph drawn by IAGA-IPM is shown in Fig 8(A), and the Q-H graph drawn by the IPM is shown in Fig 8(B). The Q-H graph is also drawn according to the two algorithms. When the three sets of parameters in different stereo ventilation network are m = 8 and n = 10, m = 82 and n = 123, and m = 117 and n = 179, their cutting conditions of rectangular blocks in the Q-H graph are represented in Fig 9.

From the test results, it can be seen that to apply IAGA-IPM to optimize Q-H graph drawing can reduce the number of rectangular block cuts. With fewer branches and nodes in the ventilation network, the topological relationship is relatively simple, so the sequence of independent paths does not change much, nor does the number of cut rectangular blocks, which cannot give full play to the advantages of IAGA-IPM. However, when the number of branches and nodes is large, its topological relationship changes in a complex way, the sequence of independent paths changes greatly, and the number of cut rectangular blocks varies greatly, which can highlight the advantages of IAGA-IPM.

## 3. Q-H graph drawing of ventilation network containing unidirectional circuits

Unidirectional circuits often exist in mines that adopt multistage fan station ventilation. This is a circuit in which the airflow directions of the branches are uniform. The performance of unidirectional circuits in mine ventilation networks is circular wind. Once the unidirectional

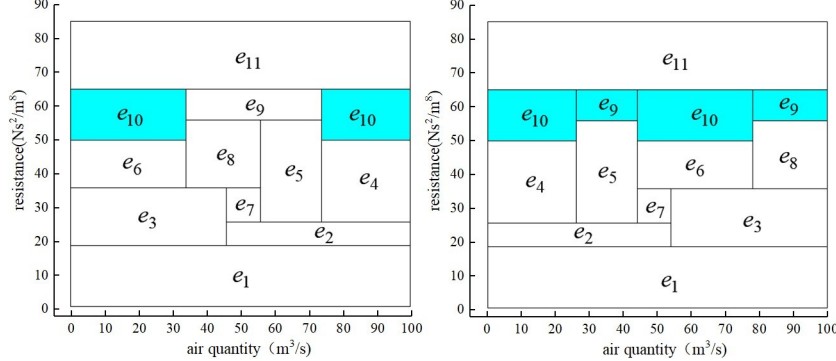

**Fig 8. Comparison of drawing effects between IAGA-IPM and IPM.** (a)IAGA-IPM drawing effect;(b)IPM drawing effect.

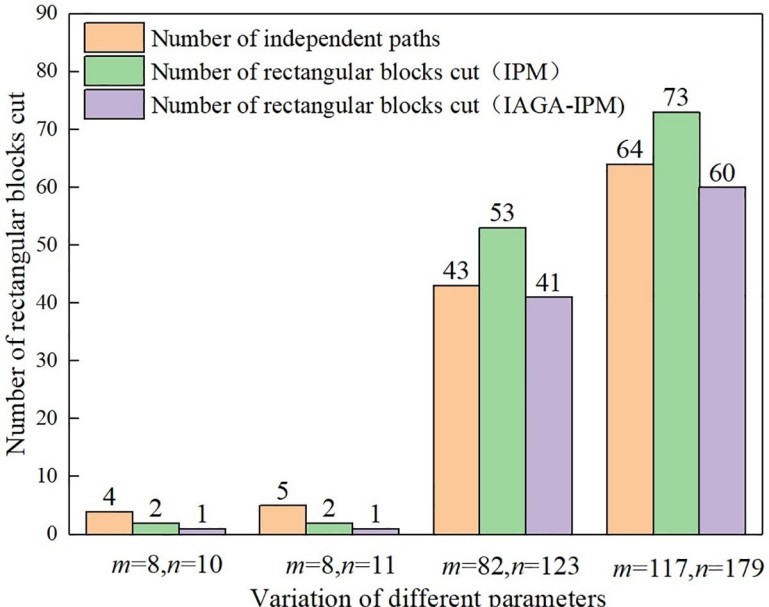

**Fig 9. Comparison of Q-H graph optimization effect.**

circuit is included in the ventilation network, all of the original path-based algorithms fail [25]. Therefore, we study the discrimination of unidirectional circuits, the improvement of the path algorithm of a ventilation network with unidirectional circuits, so as to solve the drawing of a Q-H graph with unidirectional circuits.

## 3.1 Mathematical model of identifying unidirectional circuits

To identify unidirectional circuits in a ventilation network, we propose a mathematical model of identifying unidirectional circuits based on a node adjacency matrix. For ventilation network graph $G = (V,E)$, the square matrix $A=(a_{ij})_{m \times m}$ of order $m = |V|$ is constructed, where:

$$a_{ij} = |\{e_k|e_k = (v_i, v_j) \in E\}| \tag{18}$$

Matrix A is called the node adjacency matrix of graph G. $a_{ij}$ represents the number of branches starting with $v_i$ and ending with $v_j$.

The k power of matrix A is denoted as $A^k = \left(a_{ij}^{(k)}\right)_{m \times m}$, where:

$$\left(a_{ij}^{(k)}\right) = \sum_{h=1}^{m} a_{ih}^{(k-1)} a_{hj} \tag{19}$$

$a_{ih} \cdot a_{hj} \neq 0$ if and only if $a_{ih} \neq 0$ and $a_{hj} \neq 0$, that is, nodes are all connected by branches from node $v_i$ to $v_h$ and from $v_h$ to $v_j$, so the value of $a_{ij}^{(2)}$ represents the number of paths from $v_i$ to the intermediate node $v_h$, and then to $v_j$, or to put it another way, the value of $a_{ij}^{(2)}$ represents the number of paths from $v_i$ to $v_j$ in two steps. Similarly, the value of $a_{ij}^{(k)}$ represents the number of paths from $v_i$ through step k to $v_j$, whereas $a_{ij}^{(k)} = 0$ represents the absence of such paths. Therefore, a theorem is proposed.

Theorem 1: If $\sum_{k=1}^{m-1} a_{ij}^{(k)} \neq 0$, then there is a path between nodes i and j.

From the view of graph theory, unidirectional circuits are paths whose beginning and end points coincide.

According to the definition of the unidirectional circuit, that is, a circuit with the same direction of airflow, once there is a unidirectional circuit, then there is a path that takes the beginning node $v_u$ and any end node $v_u$, and a path with overlapping beginning and end nodes is a unidirectional circuit. Therefore, as long as there is a path starting and ending with $v_i(v_i \in V)$, a unidirectional circuit is determined. Combining this with Theorem 1, if $\sum_{k=1}^{m-1} a_{ij}^{(k)} \neq 0$, that is, any element on the diagonal in $A^k = (a_{ij}^{(k)})_{m \times m}$ is not 0, then $v_i(v_i \in V)$ must be a node in the unidirectional circuit.

To sum up, a mathematical model for identifying unidirectional circuits of ventilation networks based on a node adjacency matrix is proposed.

(1) Determine the node adjacency matrix A of the preset ventilation network.

(2) Calculate $A^2, \ldots, A^{m-1}$. As the path limit length is m-1 (when all nodes are in a straight line), there is no need to calculate $A^m$.

(3) If the diagonal element in $A^k(k = 1,2,\ldots,m-1)$ has a nonzero value, a unidirectional circuit exists. Between nodes corresponding to nonzero elements on the diagonal constitute unidirectional circuits.

For the ventilation network shown in Fig 10, the process of identifying unidirectional circuits using the above mathematical model is as follows.

(1) Node adjacency matrix of ventilation network is shown in Fig 10.

$$A = \begin{bmatrix} 0 & 1 & 0 & 0 & 0 & 0 \\ 0 & 0 & 1 & 0 & 0 & 0 \\ 0 & 0 & 0 & 1 & 1 & 0 \\ 0 & 1 & 0 & 0 & 1 & 0 \\ 0 & 0 & 0 & 0 & 0 & 1 \\ 0 & 0 & 0 & 0 & 0 & 0 \end{bmatrix}$$

(2) Calculate $A^2, A^3, A^4, A^5$.

$$A^2 = \begin{bmatrix} 0 & 0 & 1 & 0 & 0 & 0 \\ 0 & 0 & 0 & 1 & 1 & 0 \\ 0 & 1 & 0 & 0 & 1 & 1 \\ 0 & 0 & 1 & 0 & 0 & 1 \\ 0 & 0 & 0 & 0 & 0 & 0 \\ 0 & 0 & 0 & 0 & 0 & 0 \end{bmatrix} \quad A^3 = \begin{bmatrix} 0 & 0 & 0 & 1 & 1 & 0 \\ 0 & 1 & 0 & 0 & 1 & 1 \\ 0 & 0 & 1 & 0 & 0 & 1 \\ 0 & 0 & 0 & 1 & 1 & 0 \\ 0 & 0 & 0 & 0 & 0 & 0 \\ 0 & 0 & 0 & 0 & 0 & 0 \end{bmatrix} \quad A^4 = \begin{bmatrix} 0 & 1 & 0 & 0 & 1 & 1 \\ 0 & 0 & 1 & 0 & 0 & 1 \\ 0 & 0 & 0 & 1 & 1 & 0 \\ 0 & 1 & 0 & 0 & 1 & 1 \\ 0 & 0 & 0 & 0 & 0 & 0 \\ 0 & 0 & 0 & 0 & 0 & 0 \end{bmatrix} \quad A^5 = \begin{bmatrix} 0 & 0 & 1 & 0 & 0 & 1 \\ 0 & 0 & 0 & 1 & 1 & 0 \\ 0 & 1 & 0 & 0 & 1 & 1 \\ 0 & 0 & 1 & 0 & 0 & 1 \\ 0 & 0 & 0 & 0 & 0 & 0 \\ 0 & 0 & 0 & 0 & 0 & 0 \end{bmatrix}$$

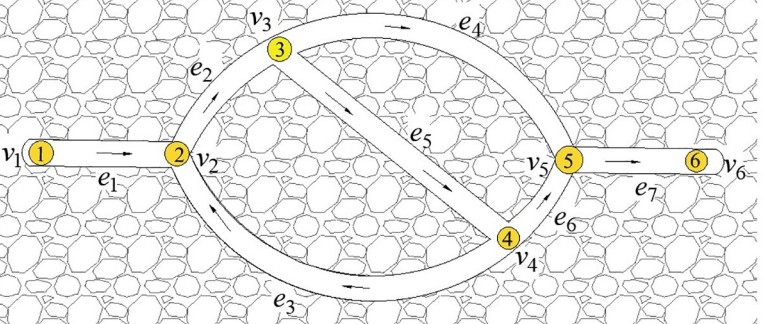

**Fig 10. A ventilation network graph with the unidirectional circuits.**

(3) It can be seen that the diagonal elements in the second, third, and fifth rows of $A^3$ are non-zero, that is, the unidirectional circuit is $\{e_2, e_5, e_3\}$.

According to the above principles, a program for identifying unidirectional circuits is written.

## 3.2 Path algorithm optimization in ventilation network with unidirectional circuits

To search the path between $v_1$ and $v_6$ in Fig 10 using the depth-first method [17] could produce the outcome $\{e_1, e_2, e_5, e_3, e_2, e_4, e_7\}$. Hence this method cannot solve the problem of finding the path of a network with unidirectional circuits, as their existence will lead to an infinite search. The search strategy must be modified to solve such a problem. Branch coloring (marking branches that have been searched) [17] remains unchanged:

$$\begin{cases} E \leftarrow E - e_k \\ E' \leftarrow E' + e_k \end{cases} \tag{20}$$

where $E = \{e_1, e_2, \ldots, e_n\}$ is the set of branches that have not been colored; $E'$ is the set of branches that have been shaded; and $e_k$ is the selected branch of the search, that is, the colored branch.

However, the edge seeking strategy [17],

$$\{e_k | e_k = (v_a, v_b) \in E, v_a \in V, v_b \in V\} \neq \phi. \tag{21}$$

is modified to:

$$\{e_k | e_k = (v_a, v_b) \in E, v_a \in V, v_b \in V, v_b \notin V(E')\} \neq \phi \tag{22}$$

where $V(E')$ is the node set corresponding to the colored branch set.

The improved depth-first algorithm has three steps to determine the whole path in the ventilation network with unidirectional circuits: (1) Calculate the source point $V^-(G)$ and rendezvous point $V^+(G)$ of the network; (2) calculate the entire path between the point pairs composed of the source point and rendezvous point; (3) sum over the paths of each pair of points.

The program chart of the path calculation between any two nodes is shown in Fig 11. The steps are as follows:

Step 1: Initialization. $i = 1$ is path number initialization.

Step 2: Seek edges. If successful, then turn to (3) for coloring. Otherwise, go to (8) to determine whether the stack is empty. If the stack is empty, then go to (9) to end the program.

Step 3: Pop down.

Step 4: Determine the source node for seeking edges.

Step 5: Determines whether the current node is the target node. If it is not the target node, then go to (2) and continue to seek edges. Otherwise, go to (7) to storage stack where the paths constituted by the branches are stored, and then add up the total number of paths.

Step 6: Identify branches that should be restored to original color.

Step 7: Add the restored branch to E.

Step 8: Determine the new source node for seeking edges.

Step 9: Quit stack.

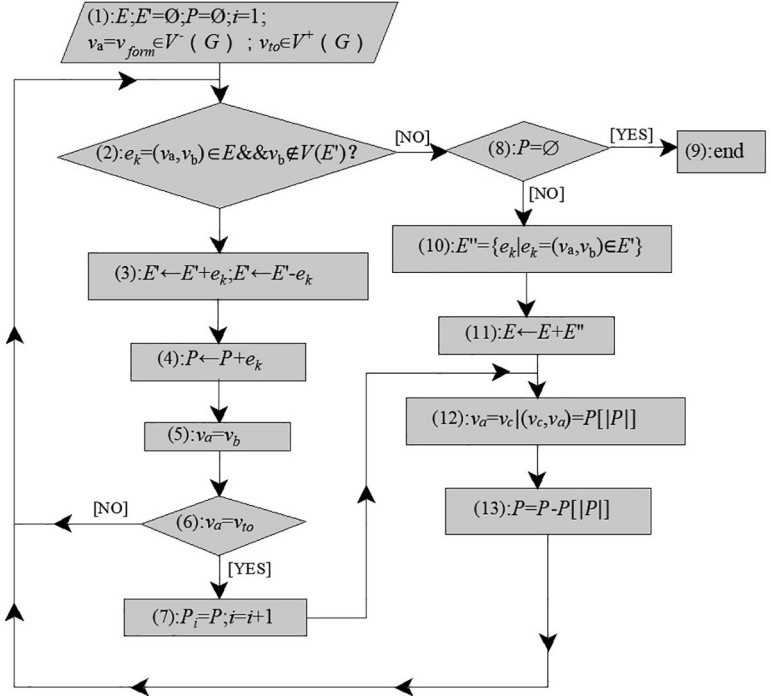

**Fig 11. Program chart of determining all paths for the network with unidirectional circuits.**

By modifying the edge seeking strategy, the depth-first algorithm can be used to find the path in the ventilation network with unidirectional circuits.

### 3.3 Q-H graph drawing method for ventilation network with unidirectional circuits

To draw a Q-H graph containing unidirectional circuits, they must be determined first. These are extracted from the ventilation network, and the network is transformed to a network without unidirectional circuits using the equivalent transformation method of the topology relation of the network, as shown below.

Suppose the ventilation network graph is $G = (V, E)$, the source point of the network is $V^-(G)$, the associated branch of this point is $E^-(G)$, the rendezvous point of the network is $V^+(G)$, the associated branch of this point is $E^+(G)$, and the fan branch that causes the unidirectional circuit is $e_f = (v_a, v_b), (v_a, v_b \in V)$. Construct a new fan branch $e'_f = (v_a, v_c), (v_c \notin V)$, and set the ventilation parameter of $e'_f$ equal to $e_f$. Then the transformed network has the following relation to the original network:

$$\begin{cases} E' = E - e_f + e'_f; \\ V' = V + v_c; \\ v_c \in V^+(G'), v_c \notin V; \\ v_b \in V^-(G') \\ V^+(G') = V^+(G) + 1; \\ V^-(G') = V^-(G) + 1. \end{cases} \quad . \tag{23}$$

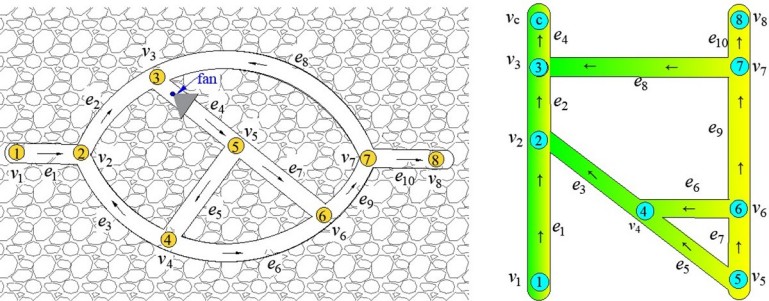

**Fig 12. The conversion of a ventilation network containing single circuit into a network without single circuit.** (a) Network graph containing unidirectional circuits.(b)Network graph after topology transformation.

It can be seen that the transformed network has the same number of branches and the same topological relation as the original network, except for the branches that cause the unidirectional circuit, whose end nodes are the source points of the new network, wheeas the end nodes of the newly constructed branches are the rendezvous points of the new network.

Taking Fig 12(A) as an example, a fan is installed on branch $e_4$ of the ventilation network. Under the action of the fan, the network forms two unidirectional circuits, $c_1 = \{e_4, e_5, e_3, e_2\}$ and $c_2 = \{e_4, e_7, e_9, e_8\}$. The air volume and resistance of the branches are shown in Table 2. The air pressure of the fan is $h_f = 100 \text{Ns}^2/\text{m}^8$, and the air quantity is $Q_f = 100 \text{m}^3/\text{s}$.

It can be seen from Table 2 that although the network is balanced, that is, it meets the laws of air quantity balance of nodes and resistance balance of circuits, the corresponding Q-H graph cannot be drawn. The reason is that, from the perspective of fan power, the nodes of highest and lowest network pressure energy are $v_5$ and $v_3$, respectively. The line of the node corresponding to $v_5$ should be at the bottom of the Q-H graph, whereas the node line corresponding to $v_3$ should be at the top. However, from the perspective of branch $e_4$, the air flows from $v_3$ to $v_5$, and the line of the node corresponding to $v_5$ should be at the top of the Q-H graph, whereas the node line corresponding to $v_3$ should be at the bottom. So, the rectangular blocks corresponding to branch $e_4$ cannot be expressed.

In Fig 12(A), the fan branch that causes the unidirectional circuit before the transformation is $e_f = (v_a, v_b) = e_4 = (v_3, v_5)$, and the newly constructed fan branch is $e'_f = e_4 = (v_3, v_c)$. The transformed network is shown in Fig 12(B), and it includes the unidirectional circuit. According to the drawing principle and method of the Q-H graph, the transformed Q-H graph is shown in Fig 13, from which it is seen that it has nine horizontal lines corresponding to nine nodes of the transformed network. The node of highest pressure energy in the entire network is node $v_5$, whose associated branches are $e_5$ and $e_7$, and the end node of branch $e_4$ is the node of lowest pressure energy.

In conclusion, it can be seen that the ventilation network with unidirectional circuits can be transformed to one without unidirectional circuits through the equivalent transformation of the topological relation in the network, and then Q-H graph drawing can be completed for the ventilation network. This method is also suitable for drawing a Q-H graph of a stereo ventilation network with unidirectional circuits.

**Table 2. Parameters of ventilation network with unidirectional circuits.**

| Branch | $e_1$ | $e_2$ | $e_3$ | $e_4$ | $e_5$ | $e_6$ | $e_7$ | $e_8$ | $e_9$ | $e_{10}$ |
|---|---|---|---|---|---|---|---|---|---|---|
| Air quantity($\text{m}^3/\text{s}$) | 5 | 65 | 60 | 100 | 40 | 20 | 60 | 35 | 40 | 5 |
| Resistance($\text{Ns}^2/\text{m}^8$) | 10 | 30 | 20 | 20 | 30 | 15 | 15 | 20 | 45 | 10 |

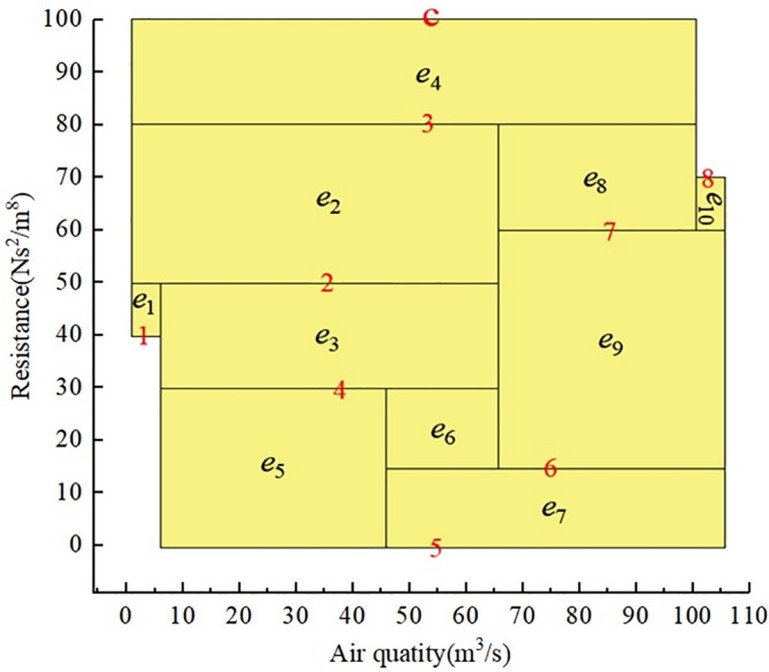

**Fig 13. Q-H graph containing unidirectional circuits.**

This paper not only solves the Q-H graph drawing problem of ventilation network with unidirectional circuits, but also improves the drawing speed compared with the drawing method in the [14]. In [14], the process of using depth-first search method to determine a path is as follows: after a path is found, the search is continued through backtracking until all branches are reached. The repeated path search process is "meaningless", which will greatly affect the efficiency of the algorithm for the network with a large number of paths. In this paper, the first path is searched through the depth-first search method, but the subsequent path is determined not by backtracking and forward search, but by directly returning to the network source point and searching again, which is a method to determine the independent path. This method can greatly reduce the search times and shorten the running time.

As [14] did not provide the corresponding algorithm source code, in the absence of unified test platform (system environment, development language), the computer running speed of the two drawing algorithms cannot be compared and analyzed in the paper, so no comparison results of specific drawing time are given. The paper only compares the two kinds of drawing speed theoretically. However, by optimizing the segmentation of rectangular blocks in the drawing process, the paper makes the drawing effect of Q-H graph more intuitive (the fewer rectangular blocks, the more intuitive). As shown in Figs 14 and 15, the Q-H graph $e_8$ and $e_9$ are divided in the method of [14], while only $e_9$ is divided in the method of this paper. It is more clear and intuitive to analyze problems.

## 4. Application of ventilation network simplification technology in Q-H graph drawing

The Q-H graph drawn according to the actual branches and nodes of the ventilation system is often too complex, with no prominent focus, which is inconvenient for analysis and research. Therefore, the ventilation network should be simplified according to the locations of key

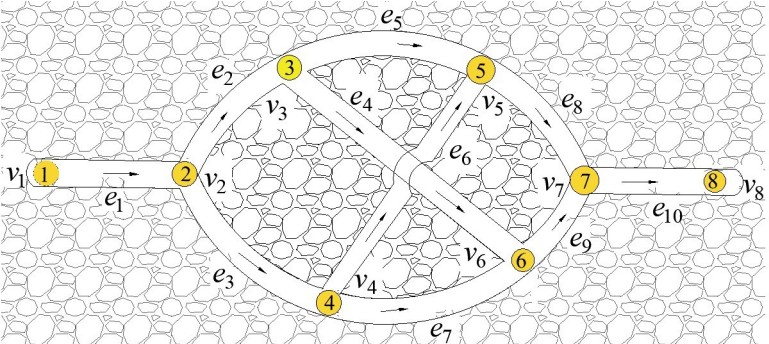

**Fig 14. Three-dimensional network graph in [14].**

nodes and the requirements of computer operation [26]. Based on graph theory and set theory, a simplified mathematical model of a ventilation network is proposed by analyzing the topological relations of the network to realize the hierarchy of the Q-H graph.

## 4.1 Mathematical model and program design of network simplification

Let $G' = (V', E')$ be the connected subgraph of network graph $G = (V, E)$, where $V, V'$ is a node set and $E, E'$ is a branch set. Then:

$$\begin{cases} |V' * V(E - E')| = 2 \\ |E'| \geq 2 \end{cases} \tag{24}$$

If the above inequality is true, then $G'$ is a subnetwork (or subnet) of network G, where $|E'|$ is the number of elements in matrix E.

The subnet must be a connected subgraph. The number of branches must be greater than or equal to 2, and the network removed subnet and the subnet must have only two rendezvous points, written as:

$$V' * V(E - E') = \{z, \bar{z}\} \tag{25}$$

According to the branch topological relations of the subnet, it is either a parallel, series, or diagonal connection. In the directed graph, $\{z, \bar{z}\}$ denotes the source and rendezvous points,

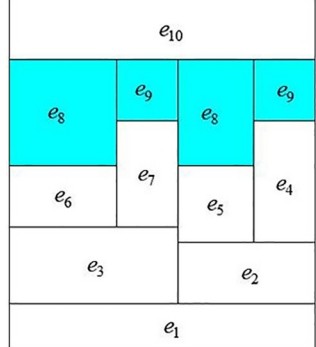
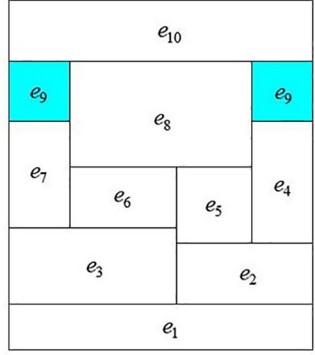

**Fig 15. Comparison between [14] and the drawing effect of this paper.** (a)Drawing effect in [14];(b)Drawing effect of this paper.

respectively, of the network, and $\{Z, \bar{Z}\}$ denotes the source and rendezvous points of the subnet. In the diagonal connection structure, it corresponds to the shunt and confluence nodes in the 7-tuples of the structure [17]. The type of subnet is determined as follows:

$$\begin{cases} |V'| = 2 (parallel) \\ |E'| = |V'| - 1 (series\,connection) \\ |E'| \neq |V'| - 1, |E'| > 5, |V'| > 4 (diagonal) \end{cases} \tag{26}$$

If $G'$ is a subnet of network G, then it can be simplified to a branch $(z, \bar{z})$ in the directed graph and a branch $\langle z, \bar{z} \rangle$ in the undirected graph. For the parallel connection, series connection and diagonal connection, the branches can be expressed as $p(z, \bar{z})$, $s(z, \bar{z})$, $d(z, \bar{z})$ and $p\langle z, \bar{z} \rangle$, $s\langle z, \bar{z} \rangle$, $d\langle z, \bar{z} \rangle$, respectively, in the directed and undirected graphs.

However, not all ventilation networks can be simplified, such as $\nabla$ and Y networks, as shown in Fig 16. Setting the clockwise direction as positive, the air quantity and resistance corresponding to the branches $\{e_1, e_2, e_3\}$ are $\{q_1, q_2, q_3\}$ and $\{r_1, r_2, r_3\}$, respectively, in the $\nabla$ network, and the air quantity and resistance corresponding to the branches $\{e_1', e_2', e_3'\}$ are $\{q_1', q_2', q_3'\}$ and $\{r_1', r_2', r_3'\}$, respectively, in the Y network. If network-equivalent parameter conversion is to be carried out, the following conditions must be met [17]:

(1) The wind volume relationship between the three nodes of $v_1$, $v_2$, and $v_3$ should satisfy:

$$\begin{cases} q_1' = q_1 + q_2; \\ q_2' = q_1 + q_3; \\ q_3' = q_2 + q_3. \end{cases} \tag{27}$$

(2) The wind pressure relationship between the three nodes of $v_1$, $v_2$, and $v_3$ should satisfy:

$$\begin{cases} r_1 q_1 |q_1| = r_1' q_1' |q_1'| + r_2' q_2' |q_2'| \\ r_2 q_2 |q_2| = r_1' q_1' |q_1'| + r_3' q_3' |q_3'| \\ r_3 q_3 |q_3| = r_2' q_2' |q_2'| + r_3' q_3' |q_3'| \end{cases} \tag{28}$$

(3) The transformed wind resistance relationship $\{r_1', r_2', r_3'\}$ still depends on $\{r_1, r_2, r_3\}$.

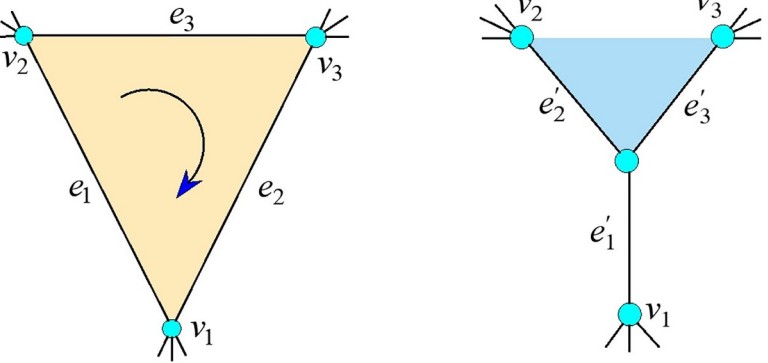

**Fig 16. Two common ventilation networks.** (a) $\nabla$ network graph; (b) Y network graph.

According to the circuit resistance balance law [17], the resistance balance formula of the $\nabla$ network is:

$$r_1 q_1 |q_1| + r_2 q_2 |q_2| + r_3 q_3 |q_3| = 0. \tag{29}$$

Formulas (27), (28), and (29) have seven equations and nine unknown numbers, so $\{r_1', r_2', r_3'\}$ has an infinite set of solutions. Hence the $\nabla$ and Y networks cannot undergo an unconditional equivalent transformation, and the equivalent wind resistance of the Y network is not only determined by the wind resistance of the $\nabla$ network, but is subject to its other parameters changes. This also is the fundamental reason why formula (24) can simplify subnets only if $|V'^* V(E-E')| = 2$. Through the above analysis, only the relatively independent network structure of one in and one out can be simplified in the ventilation network.

The simplification process of the network is hierarchical. Branches X and Y are connected in parallel, but may be in series with branch Z. After a diagonal connection subnet is simplified to a branch, the branch forms a new parallel or series relationship with other branches. This goes on until it can be simplified no more. The program chart of automatic simplification is shown in Fig 17.

In Fig 17, $E' = \{dep(z, \bar{z})\}$ in module (2) represents the branch set searched by depth-first traversal search from z to $\bar{z}$. A double line indicates that a subnet must be formed between nodes $z = V[i]$ and $\bar{z} = V[j]$, but the subnet may contain smaller subnets, so it must be treated as a network, and the network simplification program may be invoked again. If the subnet contains a smaller subnet, then the process continues until it is determined that it contains no smaller subnet, and then it can be simplified to a branch.

We use computer language to design a network simplification program. Take Fig 18(A) as an example to illustrate the hierarchy and process of network simplification. The simplification is divided into six levels, and the network is finally simplified to a branch. The process of simplification is as shown in Fig 18(B).

Layer 1: Parallel:$P(v_5, v_6) = \{e_7, e_8\}; P(v_{10}, v_{12}) = \{e_{16}, e_{17}\}$

Layer 2: Series: $S(v_5, v_7) = \{P(v_5, v_7), e_9\}$.

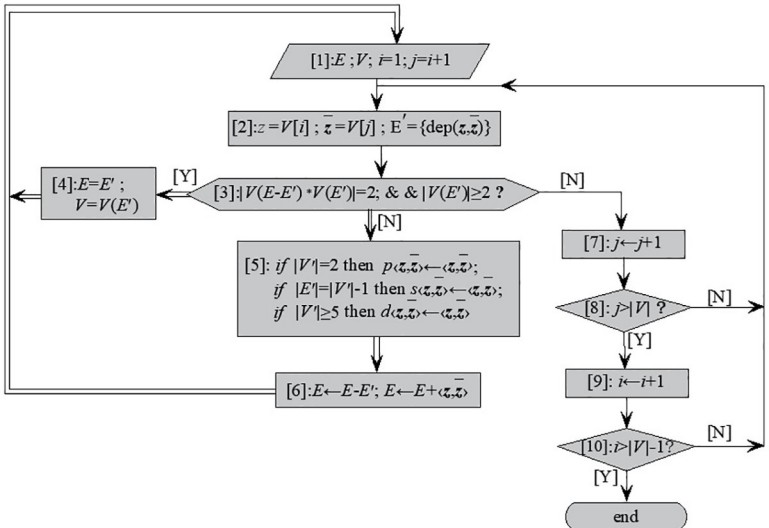

**Fig 17. Program chart of automatic simplification.**

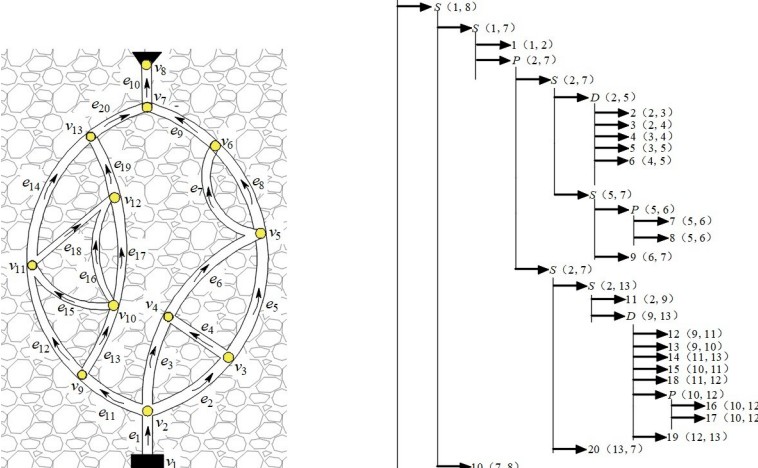

**Fig 18. Network simplification process.** (a) An example ventilation; (b) Result of network simplification.

Layer 3: Diagonal: $D(v_9,v_{13}) = \{e_{12},e_{13},e_{14},e_{15},P(v_{10},v_{12}),e_{18},e_{19}\}$, $D(v_2,v_5) = \{e_2,e_3,e_4,e_5,e_6\}$.

Layer 4: Series: $S(v_2,v_7) = \{e_{11},D(v_9,v_{13}),e_{20}\}$, $S(v_2,v_7) = \{D(v_2,v_5),S(v_5,v_7)\}$

Layer 5: Parallel: $P(v_2,v_7) = \{S(v_2,v_7),S(v_2,v_7)\}$.

Layer 6: Series: $S(v_1,v_8) = \{e_1,P(v_2,v_7),e_{10}\}$

## 4.2 Realization of Q-H graph hierarchy through network simplification

When the scale of a ventilation network is relatively large, more rectangular blocks represent branches in the Q-H graph, which is not conducive to the intuitive reflection of the network state. Network simplification technology makes it possible to realize the hierarchy of the Q-H graph. Take Fig 18(A) as an example.

The Q-H graph for Fig 18(A) is shown in Fig 19(A). It is a planar network, and there are 20 rectangular blocks, which appear to be disordered. According to the simplification hierarchy of Fig 18(B), the network is simplified to a series branch $s(v_1,v_8)$, whose corresponding Q-H graph is a rectangular block. The width of the block is the total flow of the network, whereas

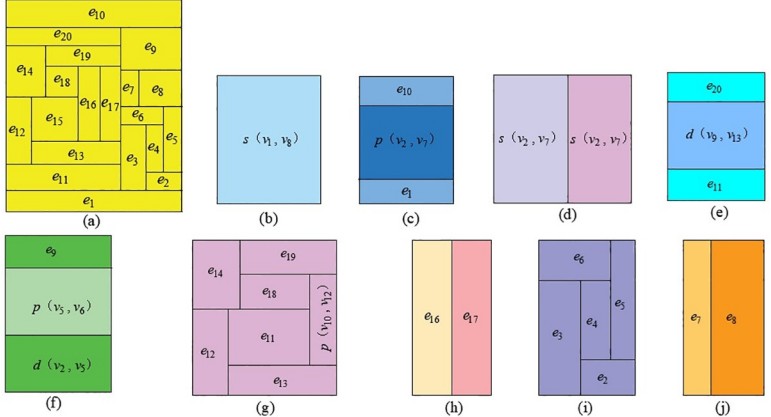

**Fig 19. The hierarchy of Q-H graph.** (a) Total Q-H graph, (b)General view, (c) $s(v_1,v_8)$, (d) $p(v_2,v_7)$, (e) $s(v_2,v_7)$ left, (f) $s(v_2,v_7)$ right, (g) $d(v_9,v_{13})$, (h) $p(v_{10},v_{12})$, (i) $d(v_2,v_5)$, (j) $p(v_5,v_6)$.

the height is the total resistance, and a quadratic curve from lower-left to upper-right is the total flow resistance. The series-simplified branch $s(v_1,v_8)$ is composed of two basic branches and a parallel-simplified branch $p(v_2,v_7)$, as shown in Fig 19(C), whereas $p(v_2,v_7)$ in Fig 19(C) is composed of two series-simplified branches $s(v_2,v_7)$, as shown in Fig 19(D). According to the simplification hierarchy of Fig 18(B), the above simplified branch $s(v_2,v_7)$ left, $s(v_2,v_7)$ right, $d(v_9,v_{13})$, $p(v_{10},v_{12})$, $d(v_2,v_5)$, $p(v_5,v_6)$ corresponding branch composition and Q-H graph are shown in Fig 19(E)–19(J), respectively.

The network simplification technology makes the Q-H graph hierarchical, which intuitively reflects the overall features of the network and describes the local situation in detail. Similarly, the ventilation network is divided into smaller subnets, and each subnet drawing is optimized by the above algorithm. A total Q-H graph is obtained by combining all subnets.

## 5. Application

The application of Q-H graph is of great value in the daily management and reconstruction of mine ventilation system. Q-H graph can be used to adjust the air quantity as needed, determine the total resistance, analyze the three-area management of mine ventilation system, map the mine air environmental parameters, map the mine pollution range and determine the escape route, etc.

### 5.1 Regulation of air volume and determination of total resistance

When the resistance and air quantity of the paths in the ventilation network are known, the amount, location, the number of paths and means of adjustment can be determined by the use of the Q-H graph. The path method is adopted to adjust (n-m+1) small resistance paths with the maximum resistance path as the benchmark,which makes the resistance of each path equal, so as to ensure the minimum power consumption. In addition, for any ventilation network, the Q-H graph can be used to determine the total resistance directly from the upper-right coordinate, without complicated calculation.

### 5.2 Management of three areas

The ventilation system can be divided into three areas: downcast air area, returning air area and using air area. The proportion of resistance of each area is an important indicator to measure the quality of the ventilation system. The proportion of resistance in three areas of natural air distribution can reflect whether the design of ventilation system is reasonable. The proportion of resistance in three areas of air distribution according to demand can reflect whether the production layout and technical management is reasonable. The Q-H graph shows the range of the three areas, as shown in Fig 20(A).

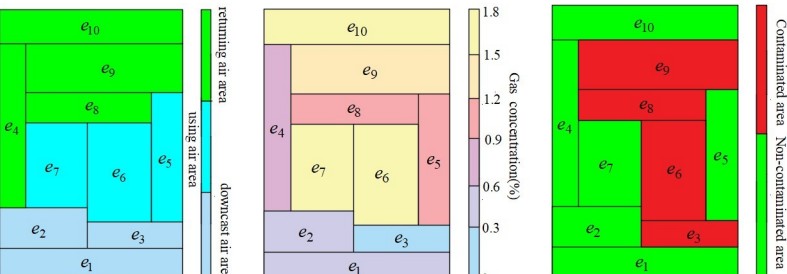

**Fig 20. Functions of Q-H graph.** (a) Q-H graph of distribution of three areas; (b)Q-H graph of distribution of gas concentration; (c) Q-H graph of distribution of pollution scope.

## 5.3 Display of mine gas concentration, temperature, humidity and other parameters

The Q-H graph can intuitively display the gas concentration, temperature and humidity of each path in the mine, which is convenient for the management of the entire mine ventilation system. Fig 20(B) is the schematic diagram of gas concentration. The gas concentration is represented by different colors in the Q-H graph, and the distribution of gas concentration in the whole mine can be clearly seen.

## 5.4 Display of mine pollution scope and escape route

The Q-H graph can show the contaminated and non-contaminated areas, as shown in the Fig 20(C). When there is a mine disaster, the Q-H graph can determine the route to escape. The workers who are blocked in the mine can avoid the disaster according to the safe route shown in the graph.

In addition to the above functions, the Q-H graph can also be applied in the following aspects: providing parameters for pressure equating fire prevention technology [12], determining boundary conditions of seepage field [12], ventilation system reconstruction [27], showing mine ventilation facilities [27], determining extreme value flow of ventilation system [28], etc. In conclusion, the Q-H graph plays a very important role in the management of mine ventilation system.

## 6. Conclusion

In this paper, independent path method, improved adaptive genetic algorithm, the equivalent transformation of the topological relation and network simplification method are used to study the modeling, optimization, specialized processing(the ventilation network with unidirectional circuits) and level-division of Q-H graph, and the following main conclusions are drawn:

An independent path method used to construct the mathematical model for drawing Q-H graphs. The improved adaptive genetic algorithm was used to optimize the cutting of the rectangular block in the process of Q-H graph drawing, whose optimal effect was realized, which is a good reference for the theoretical research and programming of the Q-H graph.

Based on node adjacency matrix theory, a mathematical model to identify unidirectional circuits was constructed. By modifying the edge-seeking strategy, the depth-first algorithm is suitable for both ordinary ventilation networks and those with unidirectional circuits. By using the equivalent transformation method of the network topological relation, the Q-H graph drawing of a network with unidirectional circuits was solved, and the drawing design theory of a complex mine ventilation network was enriched and improved.

The equivalent simplified mathematical model of the network was established, and the simplified process and principle were studied. Using network simplification technology can effectively improve the analysis speed and shorten the drawing time of a complex mine ventilation network, and realize the hierarchy of the Q-H graph, so it can intuitively reflect the overall features of the network and describe a local situation in detail.

The Q-H graph has a powerful practical function and has changed the traditional ventilation management methods. It is a new reform, which opens a new way for the modernization of ventilation management technology.

## Supporting information

**S1 File.**
(PDF)

## Acknowledgments

We thank LetPub (www.letpub.com) for its linguistic assistance during the preparation of this manuscript.

## Author Contributions

**Conceptualization:** Jinzhang Jia.

**Formal analysis:** Mingyu Wang.

**Investigation:** Dinglin Ke.

**Validation:** Yumo Wu, Dan Zhao.

**Writing – original draft:** Bin Li.

**Writing – review & editing:** Jinzhang Jia.

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
