## [Decision Letter · Decision Letter 0]

29 Sep 2020

PONE-D-20-26271

Study on O ptimization of M ine V entilation N etwork F eature G raph

PLOS ONE

Dear Dr. Bin Li

Thank you for submitting your manuscript to PLOS ONE. After careful consideration, we feel that it has merit but does not fully meet PLOS ONE’s publication criteria as it currently stands. Therefore, we invite you to submit a revised version of the manuscript that addresses the points raised during the review process:

1. Make this paper more readable by simplifying the notations used and providing an illustrative example;<o:p></o:p>

2. Compare your approach to existing approaches; and

<o:p>3. Proof-read the paper</o:p>

We look forward to receiving your revised manuscript.

Kind regards,

Xiaodi Huang, PhD

Academic Editor

PLOS ONE

Additional Editor Comments:

Title: remove the words of “Study on”

1 Introduction

What is the basic idea of your approach? Compared to the existing ones, what are the differences from your proposed approach? Give a simple example of a ventilation network. Please answer these two questions in introduction

2.1

The definition given in 2.1 is just a normal graph. What the specifical features particularly for drawing a ventilation network are?

2.2.1  remove the words of” Research on”

2.2.2

The objective function is important. Compared the existing objective functions for this problem, what the new in your function? Or the same?

Add one section for comparison (Before Section 4)

In the cover letter, you claim “The rapid and accurate drawing of the ventilation network feature graphs is significant”

Please add one section on 1. comparing your proposed method and existing approaches from at least two aspects of time (rapid) and accuracy (I don’t know how to measure the accuracy of a drawing); and 2. Providing the drawing examples for comparison

6. removed (1), (2) ….

Please proof-read your whole manuscript.

Journal Requirements:

"This research was supported by the National Natural Science Foundation of China (No. 51374121) and

funded by Liaoning Distinguished Professor (551710007007), funded project of the Liaoning Million Talents project (2019-45-15), and the Natural Science Foundation of Liaoning Province (2019-MS-162)"

"yes"

"The authors declare that they have no known competing financial interests or personal relationships that could have appeared to influence the work reported in this paper."

We note that one or more of the authors are employed by a commercial company: Power China HuaDong Engineering Corporation Limited.

3.1. Please provide an amended Funding Statement declaring this commercial affiliation, as well as a statement regarding the Role of Funders in your study. If the funding organization did not play a role in the study design, data collection and analysis, decision to publish, or preparation of the manuscript and only provided financial support in the form of authors' salaries and/or research materials, please review your statements relating to the author contributions, and ensure you have specifically and accurately indicated the role(s) that these authors had in your study. You can update author roles in the Author Contributions section of the online submission form.

3.2. Please also provide an updated Competing Interests Statement declaring this commercial affiliation along with any other relevant declarations relating to employment, consultancy, patents, products in development, or marketed products, etc.  

5. Please note that in order to use the direct billing option the corresponding author must be affiliated with the chosen institute. Please either amend your manuscript or remove this option (via Edit Submission).

Reviewers' comments:

Reviewer's Responses to Questions

**Comments to the Author**

1. Is the manuscript technically sound, and do the data support the conclusions?

Reviewer #1: Partly

Reviewer #2: Partly

2. Has the statistical analysis been performed appropriately and rigorously? 

Reviewer #1: I Don't Know

Reviewer #2: Yes

3. Have the authors made all data underlying the findings in their manuscript fully available?

Reviewer #1: Yes

Reviewer #2: Yes

4. Is the manuscript presented in an intelligible fashion and written in standard English?

Reviewer #1: No

Reviewer #2: Yes

5. Review Comments to the Author

Reviewer #1: The paper introduces an algorithm for drawing ventilation network feature graphs. The application of the ventilation network can be potentially applicable in mining.

In my option, although the technical contribution of this work is solid. However, the writing is really complex and really hard to follow, especially for any readers who are not in the field. There is no clear connection among the sections as well as the message/narrative that the authors wish to deliver throughout the paper. It is really to read the English expression in the paper as well. I would strongly recommend to improve the writing and the presentation in prior to the resubmission.

The last paragraph in the Introduction section needs to be revised to express the contribution of the paper better. And this contribution should be aligned with those in the abstract and other place(s).

There are long lists of citing works in the Introduction section without any discussion on them (e.g. [1-5], [7-9], [11-16] etc. I think it is much more useful if the authors provide at least a brief discussion or presentation on these cited works.

The authors have the whole long section 2 to explain about the drawing model and optimisation research of the Q-H Graph. This section also includes quite an extensive technical presentation. I feel that this is quite excessive that might confuse the reader(s) on the contribution of the paper and the existing work. Perhaps, it is a good idea to short this section and reference to other paper(s) for further information.

Section 5 and section Section 6 need to be re-written. It is not a good idea to use just list points for the sections in a scientific paper.

Minor points:

+ Figure 1 and Figure 2: it is a good idea explain how the planar ventilation graph can transfer to the Q-H graph with some illustration, and how the layout partitioning was generated including the position and size of the rectangles. This should be done similar for other Figure(s) if needed.

+ Figure 8: it is a good idea to link the Steps in the text after Figure 8 with the Figure. It is hard to know which steps are referring to which items in the flow charts.

+ Figure 10: I think it is a bad idea to use the 3D bar graphs. The inconsistency in bar height in 2D display can easily mislead the interpretation of the values.

+ Figure 12, page 12, at the caption, please fix the format problem for the axes labels

Reviewer #2: This paper presents a study on optimization of mine ventilation network feature graph. More specifically, it discusses theoretical research to the optimization of the Q-H graph drawing algorithm. Based on the discussion, a mathematical model based on the node adjacency matrix method for unidirectional circuit discrimination is constructed. It is claimed that the the proposed network simplification technology makes the drawing concise and can be applied to ventilation systems such as of subways, tunnels, and large shopping malls.

I am not an expert on mathematical expressions. But the paper seems to have addressed an interesting issue that can be very useful in practice. The paper is well presented and the research is well motivated.

I have a minor suggestion though. The paper should discuss in more detail about the relevant research so that we can understand better the novelty and contribution of the paper to the research body.

6. PLOS authors have the option to publish the peer review history of their article (what does this mean?). If published, this will include your full peer review and any attached files.

Reviewer #1: No

Reviewer #2: No

---

## [Author Response · Author response to Decision Letter 0]

16 Oct 2020

The response file has been uploaded separately

---

## [Decision Letter · Decision Letter 1]

26 Oct 2020

Optimization of Mine Ventilation Network Feature Graph

PONE-D-20-26271R1

Dear Dr. Bin Li

We’re pleased to inform you that your manuscript has been judged scientifically suitable for publication and will be formally accepted for publication once it meets all outstanding technical requirements.

Kind regards,

Xiaodi Huang, PhD

Academic Editor

PLOS ONE

Additional Editor Comments (optional):

Please proofread the paper.by a native English speaker.  for example, in the abstract:

"A ventilation network feature graph is the best way to directly and quantitatively reflect the features of a ventilation network..."

1. A graph is NOT a way;and 2. Reflect  represent...

Reviewers' comments:

Reviewer's Responses to Questions

**Comments to the Author**

1. If the authors have adequately addressed your comments raised in a previous round of review and you feel that this manuscript is now acceptable for publication, you may indicate that here to bypass the “Comments to the Author” section, enter your conflict of interest statement in the “Confidential to Editor” section, and submit your "Accept" recommendation.

Reviewer #1: All comments have been addressed

Reviewer #2: All comments have been addressed

2. Is the manuscript technically sound, and do the data support the conclusions?

Reviewer #1: Partly

Reviewer #2: Partly

3. Has the statistical analysis been performed appropriately and rigorously? 

Reviewer #1: Yes

Reviewer #2: N/A

4. Have the authors made all data underlying the findings in their manuscript fully available?

Reviewer #1: Yes

Reviewer #2: No

5. Is the manuscript presented in an intelligible fashion and written in standard English?

Reviewer #1: Yes

Reviewer #2: Yes

6. Review Comments to the Author

Reviewer #1: The authors has addressed the my comments. This version is much clearer and improved compared to the original one.

Reviewer #2: The authors have made some improvements for the paper based on reviewers comments. And I thus support Accept

7. PLOS authors have the option to publish the peer review history of their article (what does this mean?). If published, this will include your full peer review and any attached files.

Reviewer #1: No

Reviewer #2: No

---

## [Editor Report · Acceptance letter]

28 Oct 2020

PONE-D-20-26271R1 

Optimization of Mine Ventilation Network Feature Graph 

Dear Dr. Li:

I'm pleased to inform you that your manuscript has been deemed suitable for publication in PLOS ONE. Congratulations! Your manuscript is now with our production department. 

Kind regards, 

on behalf of

Dr. Xiaodi Huang 

Academic Editor

PLOS ONE